# Synthesis of Nanosilica via Olivine Mineral Carbonation under High Pressure in an Autoclave

**Srecko Stopic [1],\*, Christian Dertmann [1], Ichiro Koiwa [2], Dario Kremer [3], Hermann Wotruba [3], Simon Etzold [4], Rainer Telle [4], Pol Knops [5] and Bernd Friedrich [1]** 

1   IME Process Metallurgy and Metal Recycling, RWTH Aachen University, Intzestrasse 3,
    52056 Aachen, Germany; cdertmann@ime-aachen.de (C.D.); bfriedrich@ime-aachen.de (B.F.)
2   Department of Applied Chemistry, College of Science and Engineering, Kanto Gakuin University,
    1-50-1, Mutsuurahigashi, Kanazawa-ku, Yokohama 236-8501, Japan; koiwa@kanto-gakuin.ac.jp
3   AMR Unit of Mineral Processing, RWTH Aachen University, Lochnerstrasse 4-20, 52064 Aachen, Germany;
    kremer@amr.rwth-aachen.de (D.K.); wotruba@amr.rwth-aachen.de (H.W.)
4   Department of Ceramics and Refractory Materials, GHI Institute of Mineral Engineering,
    RWTH Aachen University, Mauerstrasse 5, 52064 Aachen, Germany; etzold@ghi.rwth-aachen.de (S.E.);
    Telle@ghi.rwth-aachen.de (R.T.)
5   Green Minerals, Rijksstraatweg 128, NL 7391 MG Twello, The Netherlands; pol@green-minerals.nl
\*   Correspondence: sstopic@ime-aachen.de

**Abstract:** Silicon dioxide nanoparticles, also known as silica nanoparticles or nanosilica, are the basis for a great deal of biomedical and catalytic research due to their stability, low toxicity and ability to be functionalized with a range of molecules and polymers. A novel synthesis route is based on $CO_2$ absorption/sequestration in an autoclave by forsterite ($Mg_2SiO_4$), which is part of the mineral group of olivines. Therefore, it is a feasible and safe method to bind carbon dioxide in carbonate compounds such as magnesite forming at the same time as the spherical particles of silica. Indifference to traditional methods of synthesis of nanosilica such as sol gel, ultrasonic spray pyrolysis method and hydrothermal synthesis using some acids and alkaline solutions, this synthesis method takes place in water solution at 175 °C and above 100 bar. Our first experiments have studied the influence of some additives such as sodium bicarbonate, oxalic acid and ascorbic acid, solid/liquid ratio and particle size on the carbonation efficiency, without any consideration of formed silica. This paper focuses on a carbonation mechanism for synthesis of nanosilica under high pressure and high temperature in an autoclave, its morphological characteristics and important parameters for silica precipitation such as pH-value and rotating speed.

**Keywords:** silica; synthesis; olivine carbonation; autoclave; precipitation

## 1. Introduction

As a result of the high nickel production costs associated with traditional pyrometallurgical techniques and the depletion of high-grade sulfide ores, renewed interest has developed concern on the production of nickel and cobalt by high pressure acid leaching (PAL) of nickel laterites. More than one third of the world's nickel is nowadays produced from laterite ores [1,2]. Laterites account for two thirds of the world's nickel resources. It is therefore likely that increasing amounts of nickel will be produced from laterites. Since laterite type ores naturally occur close to the surface, economical open pit mining techniques are employed to recover the ore after removal of the overburden [3]. The laterite ore consists of fresh saprolite such as $K_{0.4}(Si_{3.0}Al_{1.0})_{4.0}(Al_{2.0}Mg_{0.3})_{2.33}O_{10}(OH)_2$ and nontronite such as $Na_{0.3}(Fe^{3+})_2(Si,Al)_4O_{10}(OH)_2 \cdot nH_2O$. These silicate ores represent the various layers in the laterite bedrock. The limonite consists mainly of goethite. This continues to a nontronite rich zone. Saprolite

is the next layer, which is distinguished from its rich magnesium silicate content. The lateritic ore mostly has a low level of Ni (1–3%), Co (max. 0.1%), Fe (20–30%) and high level of $SiO_2$ (more than 50%). The treatment of silicate based ores with different acids under an atmospheric pressure leads to formation of silica gel and breaking of leaching process. Silica gel represents an amorphous and porous form of silicon dioxide consisting of an irregular tridimensional framework of alternating silicon and oxygen atoms with nanometer-scale pores [4].

Similarly, high Si content in red mud and its slags produced by pyrometallurgical treatment for the Fe removal makes these secondary resources untreatable with conventional acid leaching routes due to the formation of silica gel. Alkan et al. [5] studied red mud and slags synthesized by electric arc furnace smelting, which contain rather moderate and extensive $SiO_2$. In the next step, the formed slag was exposed separately by red mud to dry digestion with sulfuric acid at room temperature aiming at selective Sc recovery without Ti and Si dissolution. An empirical dry digestion-leaching model was proposed for each starting material in a comparative manner in order to prevent the formation of silica gel using sulfuric acid.

The Eudialyte concentrate is a potential rare earth elements (REE) primary resource due to its good solubility in acid, low radioactivity and relatively high REE content (about 2%), but also contains more than 50% of silica. The treatment of the Eudialyte concentrate can produce silica gel during a treatment with some acids [6]. The main challenge is avoiding the formation of silica gel, which is non-filterable when using acid to extract REE. Ma et al. [7] have studied neural network modeling for the optimization of the extraction of rare earth elements from the Eudialyte concentrate by dry digestion and a subsequent leaching avoiding the formation of silica gel in the presence of the hydrochloric acid.

Development of ceramic nanoparticles such as silica, alumina and titania with improved properties has been studied with much success in several areas such as synthesis and surface science [8,9]. Advancement in nanotechnology has led to the production of nanosized silica, which has been widely used as filler in catalysis and glass industry. The silica particles extracted from natural resources contains metal impurities and are not favorable for advanced scientific and industrial applications.

The sol-gel process is widely applied to produce silica, glass, and ceramic materials due to its ability to form pure and homogenous products at mild conditions. The process involves hydrolysis and condensation of metal alkoxides ($Si(OR)_4$) such as tetraethylorthosilicate (TEOS, $Si(OC_2H_5)_4$) or inorganic salts such as sodium silicate ($Na_2SiO_3$) in the presence of mineral acid (e.g., HCl) or base (e.g., $NH_3$) as catalyst [10]. The synthesis of spherical hollow silica particles from sodium silicate solution with boric acid or urea as an additive was carried out by the ultrasonic spray pyrolysis method. This work dealt with the effect of four parameters (the concentration of the boric acid and urea, feed rate of reactant, reaction temperature and time) on particle size and standard deviation. As a result, the mean particle size and standard deviation decreased with increasing of all parameters except urea [11]. Ratanathavorn et al. [12] have studied silica nanoparticles synthesis by ultrasonic spray pyrolysis (USP) technique using tetraethylorthosilicate (TEOS) as a precursor in order to produce a fixative material for cream perfume fomulation. The results showed that the synthesis temperature of 500 °C provided the smallest size of silica nanoparticle, about 106 nm. The particle size decreased from 347 nm to 106 nm when the synthesis temperature increased from 300 °C to 500 °C.

The ultra-small hollow silica nanoparticles were synthesized using the prepared amorphous calcium carbonate (ACC) particles as a template. The ACC particles were firstly prepared by the carbonation method, which the procedure was conducted in the methanol solvent to form the $Ca(OCH_3)_2$ layers on the ACC particles. An effect of methanol concentration on the morphology of ACC particles was also investigated [13]. ACC particles were prepared by a carbonation method via bubbling $CO_2$ gas into calcium ions dispersing in methanol solution. An effect of methanol concentration on the $CaCO_3$ formation was investigated. The pH of the ACC preparation was studied in a range of 9.4 and 10. After that, ultra-small HSNPs were synthesized using the prepared ACC particles in the one-pot process. The results suggested that the synthesis of HSNPs using the ultra-small

ACC particles via the one-pot process is one of the most effective methods to produce ultra-small HSNP regarding to save energy and cost.

In a mineral carbonation process, silicate minerals can also be used as feedstock to form carbonates and $H_4SiO_4$, that are chemically stable in a geological timeframe. Silicate minerals usually are richer in alkaline earth metal content such as magnesium, sodium, and calcium. Common silicate minerals suitable for carbonation are forsterite ($Mg_2SiO_4$), antigorite ($Mg_3Si_2O_5(OH)_4$) and wollastonite ($CaSiO_3$) and their overall reaction rates are given in Equations (1)–(3):

$$Mg_2SiO_4(s) + 2CO_2(g) + 2H_2O(l) = 2MgCO_3(s) + H_4SiO_4(aq) + 89 \text{ kJ/mol} \tag{1}$$

$$Mg_3Si_2O_5(OH)_4(s) + 3CO_2(g) + 2H_2O(l) = 3MgCO_3(s) + 2H_4SiO_4(aq) + 64 \text{ kJ/mol} \tag{2}$$

$$CaSiO_3(s) + CO_2(g) + 2H_2O(l) = CaCO_3(s) + H_4SiO_4(aq) + 90 \text{ kJ/mol} \tag{3}$$

Stopic et al. [14] have shown the reaction path of direct forsterite carbonation in the aqueous solution without any deeper consideration of the formed silica particles as shown Equations (1)–(3) and Figure 1.

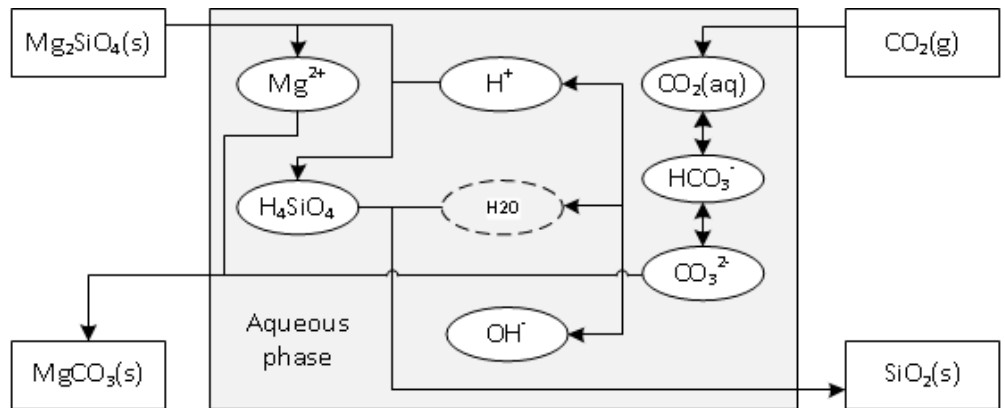

**Figure 1.** Reaction path of direct forsterite carbonation in the aqueous solution.

Although olivine is one mixed crystalline material (Mg, Fe)$_2$SiO$_4$, for simplicity, olivine consists only of $Mg_2SiO_4$, namely forsterite. First, gaseous carbon dioxide dissolves in the aqueous solution. Simultaneously, forsterite is dissolved in the aqueous solution (Equation (4)) forming aqueous silicic acid, then precipitates as amorphous silica (Equation (5)), which is a by-product, and lastly magnesium ions and carbonate form magnesite as shown with Equation (6):

$$Mg_2SiO_4(s) + 4\,H^+(aq) \xrightarrow{r_{Mg_2SiO_4}} 2\,Mg^{2+}(aq) + H_4SiO_4(aq) \tag{4}$$

$$H_4SiO_4(aq) \xrightarrow{r_{SiO_2}} SiO_2(s) + 2\,H_2O(l) \tag{5}$$

$$Mg^{2+}(aq) + CO_3^{2-}(aq) \xrightarrow{r_{MgCO_3}} MgCO_3(s) \tag{6}$$

The determination of process parameters such as temperature, pressure and pH for maximum overall conversion rates is elementary. Direct $CO_2$ sequestration at high pressure with olivine as a feedstock has already been performed in numerous studies at different temperatures and pressures with or without the use of additives such as carboxylic acid, and sodium hydroxide. It is reported that optimal reaction conditions are in the temperature range of 150–185 °C and in the pressure range of 135–150 bar [15]. Additives are reported to have a positive influence on carbonation rate, but without a study in detail. Optimal addition of additives are reported by Bearat et al. [16] in studies about the mechanism that limits aqueous olivine carbonation reactivity under the optimum sequestration reaction conditions observed as follows: 1 M NaCl + 0.64 M NaHCO$_3$, at 185 °C and P

($CO_2$) about 135 bar. A reaction limiting silica-rich passivating layer forms on the feedstock grains, slowing down carbonate formation and raising process costs. Eikeland [17] reported that NaCl does not have significant influence on carbonation rate. The presented results show a conversion rate of more than 90% using a $NaHCO_3$ concentration of 0.5 M, without adding NaCl. Ideally, the solid phases exist as pure phases without growing together. In reality, different observations are made on the behavior of solid phases. Daval et al. [18] reported about the high influence of amorphous silica layer formation on the dissolution rate of olivine at 90 °C and elevated pressure of carbon dioxide. This passivating layer may be either built up from non-stoichiometric dissolution, precipitation of amorphous silica on forsterite particles or a combination of both. These previously mentioned results suggest that the formation of amorphous silica layers plays an important role in controlling the rate of olivine dissolution by passivating the surface of olivine, an effect that has yet to be quantified and incorporated into standard reactive-transport codes. In contrast to that, Oelkers et al. [19] and Hänchen [20] observed stoichiometric dissolution and no build-up of a passivating layer except during start-up of experiments. Furthermore, magnesite may precipitate on undissolved forsterite particles leading to a surface area reduction and therefore a reduction on forsterite dissolution rate, which was reported by Turri et al. [21]. In addition to this undesired intermixing of solids, they observed pure particles of magnesite to be predominant in the intermediate particle class, amorphous silica particles to be mainly present in the smallest particle class and unreacted olivine particles to be predominant in the largest particle class. This knowledge may be of value for subsequent separation of products such as magnesium carbonate and silica.

$CO_2$ sequestration with olivine as a feedstock was performed in a rocking batch autoclave at 175 °C and 100 bars in an aqueous solution and a $CO_2$-rich gas phase from 0.5 to 12 h. Turri et al. [21] showed maintainable recovery of separate fractions of silica, carbonates and unreacted olivine. Characterization of the recovered solids revealed that carbonates predominate in particle size range below 40 μm. The larger, residue fraction of the final product after carbonation consisted mainly of unreacted olivine, while silica is more present in the form of very fine spherical particles. An addition of sodium hydrogen carbonate at 0.64 M, oxalic acid at 0.5 M and ascorbic acid at 0.01 M was successfully applied in order to obtain maximal carbonation, what leads also to a complete formation of silica.

Our paper deals with the formation of magnesium carbonate and especially nanosilica using an olivine from Norway (40.1 MgO, and 48.7 wt % $SiO_2$) and with special attention on morphological characteristics of the obtained product and water solution after filtration, which was determined by structure and composition analysis (XRD, SEM; EDS; TEM and STEM).

## 2. Experimental Section

### 2.1. Materials

The samples used represent Steinsvik olivine from Norway as analyzed by a PW2404 XRF device (Malvern Panalytical B.V., Eindhoven, The Netherlands) and as shown in Table 1.

**Table 1.** Chemical composition of the investigated olivine from Norway (fraction between 20 and 63 μm) as analyzed by X-ray fluorescence (XRF) in wt %.

| Component | $SiO_2$ | $Al_2O_3$ | $Fe_2O_3$ | CaO | MgO | $K_2O$ | MnO | $Cr_2O_3$ | ZnO | NiO |
|---|---|---|---|---|---|---|---|---|---|---|
| in wt % | 48.7 | 0.5 | 7.8 | 0.2 | 41.0 | 0.1 | 0.1 | 0.4 | 0.1 | 1.2 |

### 2.2. Procedures

The treatment of olivine was performed using the operations such as milling, sieving, carbonation in an autoclave, filtration and chemical analysis of solid and liquid sample shown at Figures 2 and 3. According to Reference 21 (Turri et al.) and Reference 14 (Stopic et al.) the carbonation tests have been carried out in the 1500 mL autoclave from Büchi Kiloclave Type 3E, Switzerland (as shown at

Figure 4) at 175 °C with 117 bar pure grade $CO_2$ in the presence and the absence of the additives such as sodium bicarbonate, oxalic acid and ascorbic acid in duration of 2–4 h. An amount ranging from 100 to 300 g sample has been added to 1000 mL solution with mixing rate 600 revolution per min in different experiments. After reaction, the liquid had very low contents of metal cations and was analyzed via the induced coupled plasma optical emission spectrometry ICP OES analysis (SPECTRO ARCOS, SPECTRO Analytical Instruments GmbH, Kleve, Germany). Characterization of the solid products was restricted to the X-ray powder diffraction XRD (Bruker AXS, Karlsruhe, Germany) and X-ray fluorescence XRF analyses using Device PW2404 (Malvern Panalytical B.V., Eindhoven, The Netherlands).

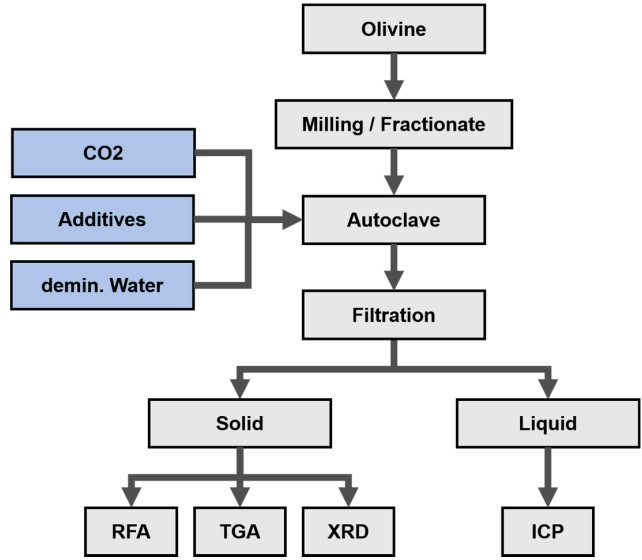

**Figure 2.** Procedure for experimental work and characterization of samples.

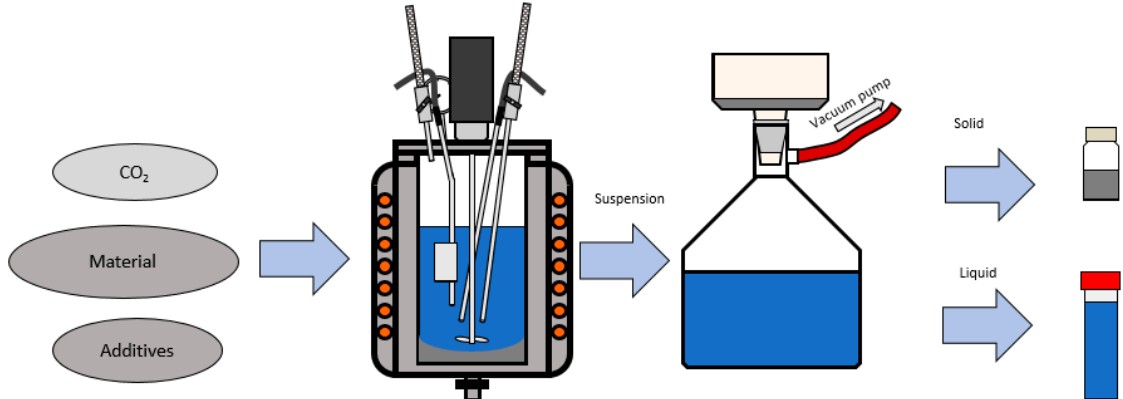

**Figure 3.** Carbonation process of olivine and sampling.

After milling the particle size fraction 20–63 µm was tested in an autoclave. The change of temperature and pressure during a heating from room temperature to 175 °C and a subsequent carbonation of 100 g olivine in 1000 mL water was followed in time. Carbonation was performed by injection of carbon dioxide from a bottle at the fixed temperature within 2 h. After this reaction time, the pressure was decreased to the atmospheric values and the solution was cooled to room temperature. After opening the cover of the autoclave the solution was filtrated as shown by Figures 2 and 3. Subsequent to drying of the solid residue, XRD analysis was performed for the product and an initial sample of olivine. X-ray powder diffraction patterns were collected by a Bruker-AXS D8 Advance diffractometer in Bragg–Brentano geometry, equipped with a copper tube coupled with a primary nickel filter providing Cu Kα1,2 radiation and LynxEye detector. The microstructure of the

solid samples was examined using a scanning electron microscope (SEM)–JEOL6380 LV (JEOL Ltd., Tokyo, Japan). Energy dispersive X-Ray spectroscopy (EDS) was utilized by JSM-6000 (JEOL Ltd., Tokyo, Japan) to reveal elemental composition of the samples analyzed by SEM. The characterization of solid and liquid products was performed using the ICP-OES analysis (SPECTRO ARCOS, SPECTRO Analytical Instruments GmbH, Kleve, Germany).

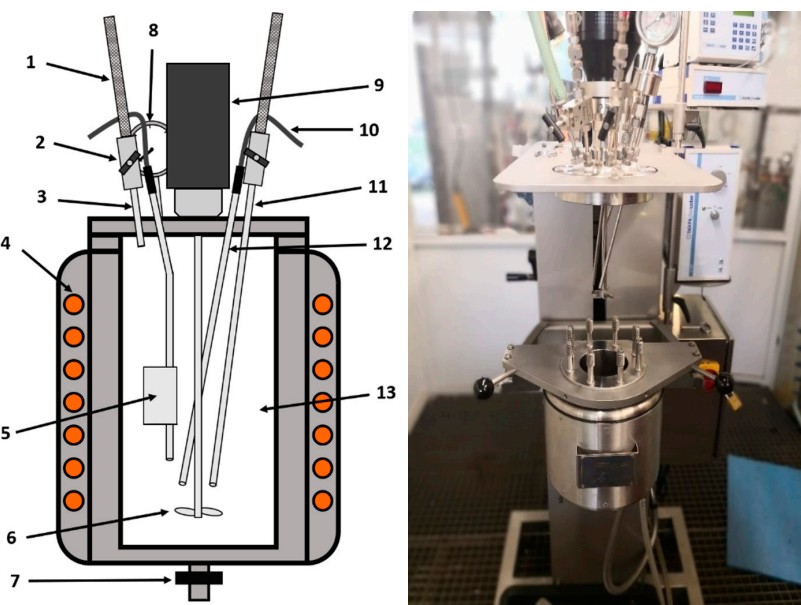

**Figure 4.** Sketch and picture of the autoclave: 1. Pressure pipe (Stahlflex); 2. needle valve; 3. tube for gas exhaust; 4. reactor shell with heating and cooling; 5. temperature sample head; 6. propeller mixer; 7. outlet valve; 8. analog manometer; 9. motor for the magnetic coupled stirrer; 10. cable for the measurement cutting site; 11. gas inlet; 12. testing rode for pressure; 13. working volume.

The values of temperature and pressure during carbonation are presented at Figure 5.

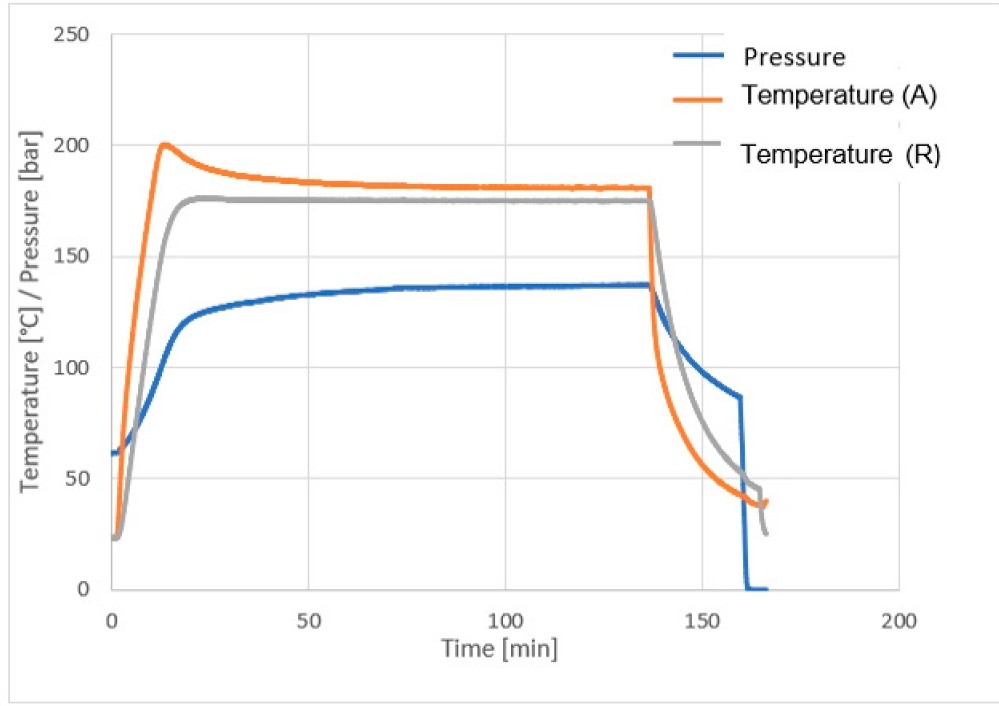

**Figure 5.** Pressure, temperature and time curves during work in autoclave (R-reaction; A-autoclave).

## 3. Results and Discussion

### 3.1. Product Characterization–Analysis via XRD and SEM of Solid Product after Carbonation

To evaluate the overall capability of the carbonation process, an experiment was performed using Norwegian olivine (20–63 µm) at 175 °C, 120 bar, 120 min, 600 rpm, in the presence of additives of sodium carbonate, oxalic and ascorbic acid considering the present mineralogical phases detected via XRD before and after the carbonation (Figure 6). A measurement range from 5–90° 2θ in 0.02° steps at 2 s per step are the chosen XRD parameters. XRD analysis was combined with a subsequent semi-quantitative evaluation of the mineral phase fractions. Table 2 provides information about the main mineral phase fractions as detected in the olivine samples within the given accuracy range. Forsterite, enstatite, lizardite and talc exhibited the most considerable mineralogical phase amounts. A decreased content of forsterite confirmed that the carbonation was successfully performed, what is indicated by 20–25% of magnesite in structure. The content of enstatite, lizardite and talc was not significantly changed, pointing out that these mineralogical phases show less reactivity within the performed process than forsterite.

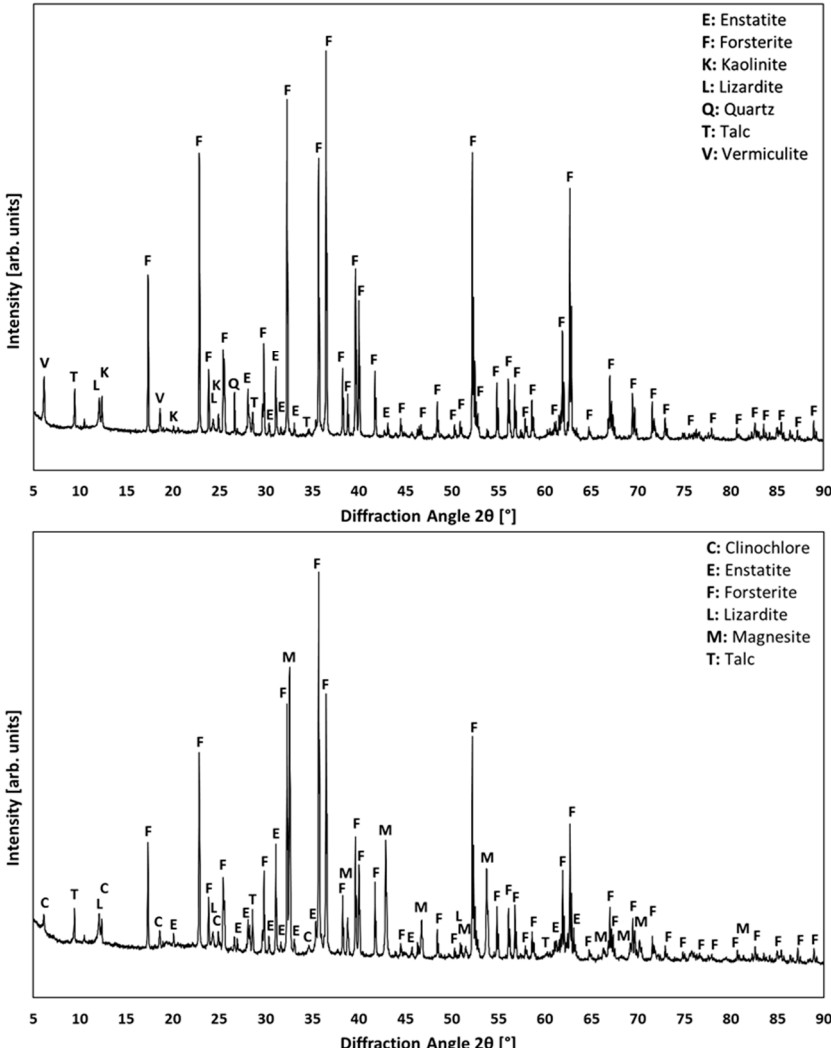

**Figure 6.** XRD analysis of Norwegian olivine samples; initial state (top) and carbonation product (bottom).

**Table 2.** The semi-quantitative XRD analysis before and after carbonation.

| Mineral Phases | Olivine, Norway (20–63 µm) | |
| --- | --- | --- |
| | Semi-Quantitative Composition in wt % | |
| | Before Carbonation | After Carbonation |
| Enstatite | 5–10 | 5–10 |
| Forsterite | 75–80 | 50–55 |
| Lizardite | ≤5 | ≤5 |
| Kaolinite | ≤5 | - |
| Talc | ≤5 | ≤5 |
| Magnesite | - | 20–25 |

The presence of magnesite and silica was confirmed using the SEM analysis of the solid product, as shown in Figure 7.

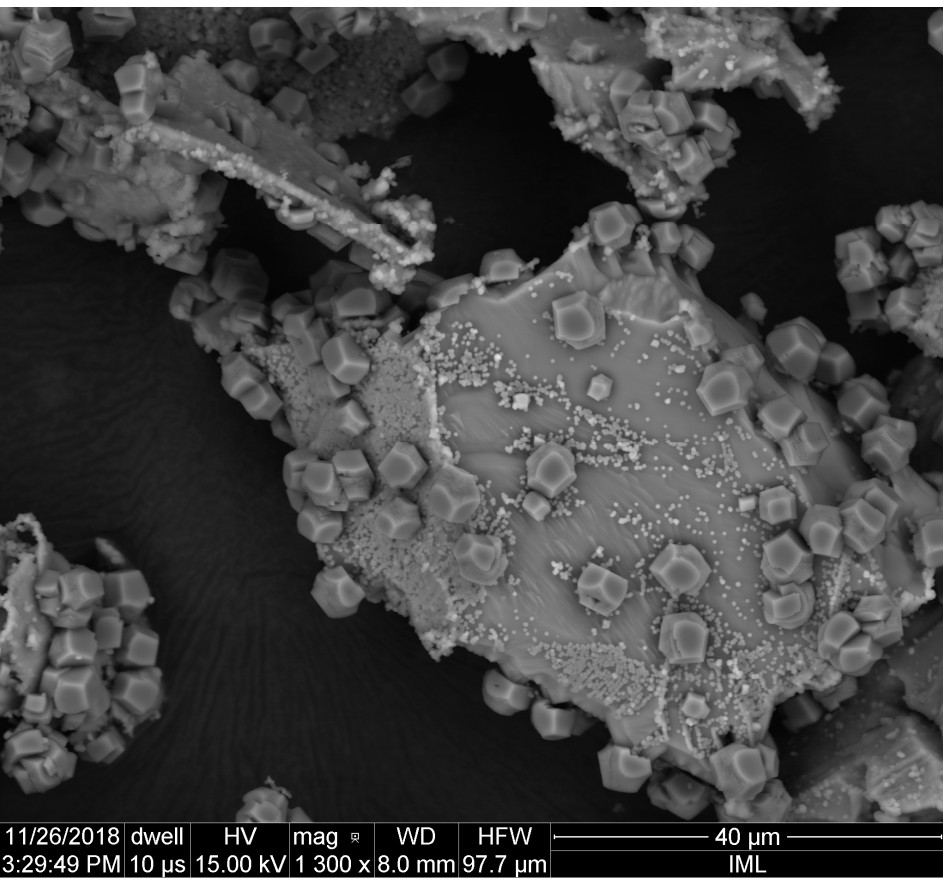

| 11/26/2018 | dwell | HV | mag ⊠ | WD | HFW | 40 µm |
| 3:29:49 PM | 10 µs | 15.00 kV | 1 300 x | 8.0 mm | 97.7 µm | IML |

**Figure 7.** SEM analysis of initial olivine sample after carbonation.

As illustrated by Figure 7, SEM-analysis has confirmed that very small particles of $SiO_2$ and magnesite are formed as rhombohedrons or hexagonal prisms at the surface of partially carbonated magnesium silicate. The challenge of future work is a separation of the formed nanosilica particles from the product.

### 3.2. Product Characterization–Analysis of Precipitate from a Water

After filtration of the carbonated products, the white and yellow precipitate appeared during staying after 7–10 days, as shown in Figure 8. Then, this precipitate was separated from water solution

and dried at 110 °C after the night. The obtained dried products were analyzed by XRD, SEM, EDS, and scanning transmission electron microscopy (STEM) by JEM-2100F (JEOL Ltd., Tokyo, Japan).

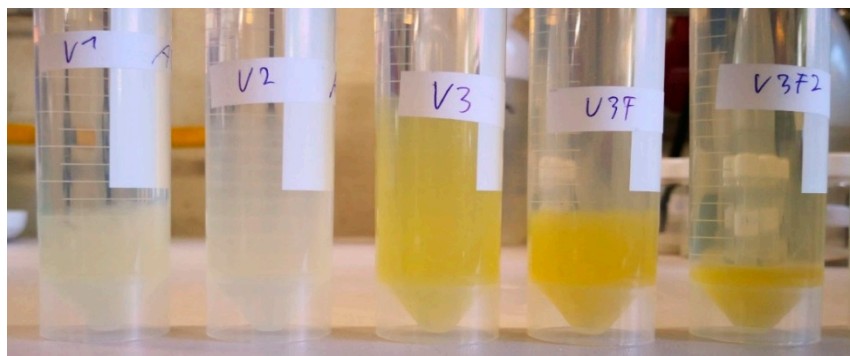

**Figure 8.** Precipitated solid residues from water solution after carbonation at 175 °C, 120 bar, s/L 1:10 (V1-600 rpm, 120 min; V2-600 rpm, 240 min; V3-1800 rpm, 120 min; V3F-1800 rpm, 40 min; V3F2-1800 rpm, 2 min).

As shown in Figure 9, an increase of stirring speed from 600 rpm to 1800 rpm in an autoclave leads to increased formation of products changing color from a white to yellow one. At the same way an increased reaction time leads to an increased production of solid residue. We suppose that an increased stirring speed has a positive influence for the separation of a formed silica rich layer at a non-reacted magnesium silicate. At the other side the pressure in an autoclave was increased from 120 to 170 bar with an increasing stirring speed from 600 to 1800 rpm, what is an additional support for the silica separation and precipitation from solution.

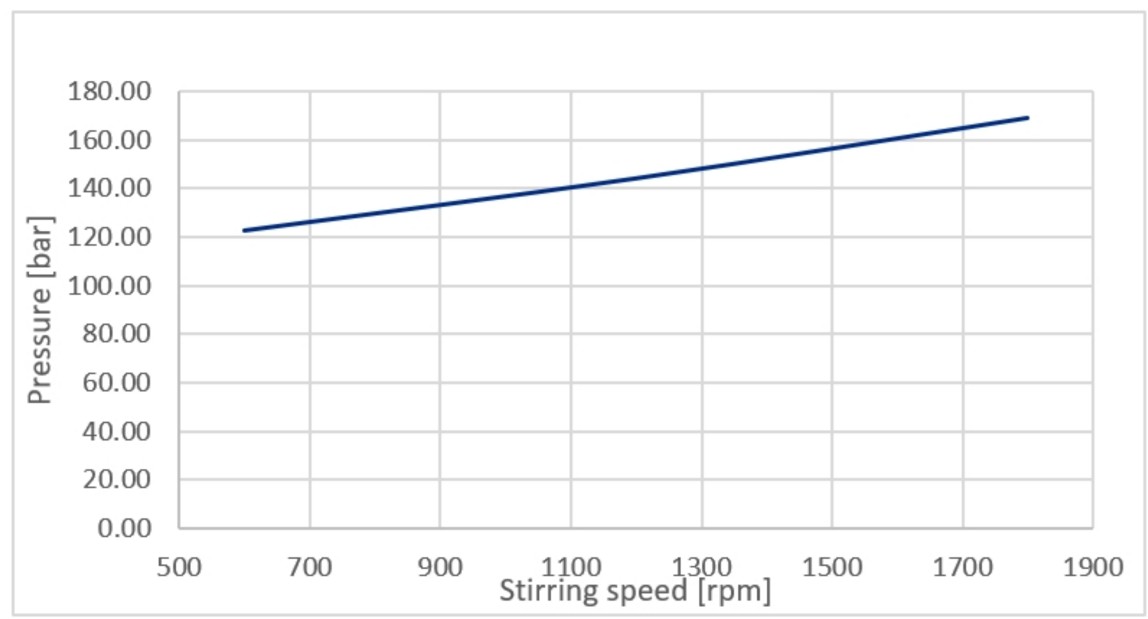

**Figure 9.** Relationship between operational pressure in an autoclave and stirring speed.

As shown in Table 3 and with Equation (7), the consumption of hydrogen ions, pH-Value was increased from the starting value 7.2 to maximal value of 8.57, forming the precipitate of $SiO_2$.

**Table 3.** The change of pH-Value of solution and pressure during a heating and carbonation.

| Experiments | pH$_{Solution}$ | pH$_{End}$ | P$_{start for heating}$ to 175 °C [bar] | P$_{Start for reaction}$ [bar] | Stirring Speed [rpm] | P$_{End}$ [bar] |
|---|---|---|---|---|---|---|
| V1 (120 min) | 7.20 | 8.32 | 61.70 | 121.60 | 600 | 137.20 |
| V2 (120 min) | 7.20 | 8.27 | 59.80 | 131.50 | 600 | 154.90 |
| V3 (120 min)) | 7.20 | 8.57 | 59.60 | 162.80 | 1800 | 170.10 |
| V3F (40 min) | 7.20 | 8.38 | 62.60 | 152.70 | 1800 | 125.60 |
| V3F2 (2 min) | 7.20 | 8.35 | 62.40 | 159.70 | 1800 | 156.60 |

$$Mg_{1.8}Fe_{0.2}SiO_4 + H^+ \rightarrow 1.8\ Mg^{2+} + 0.2\ Fe^{2+} + SiO_2 + H_2O \tag{7}$$

The maximal pressure amounted 162.2–170.1 bar at 1800 rpm, what leads to an increased separation of nanosilica. Oelkers et al. [22] studied the products after carbonation of olivine via the reaction (8). They found that quartz is not stable at partial $CO_2$ pressures between 21.4 and 223.8 bar at temperatures from 120 to 200 °C. The spherical particles are containing high amounts of silicon and oxide atoms, which is confirmed by the EDS analysis (Figure 9).

$$0.5\ Mg_2SiO_4 + CO_2 \rightarrow MgCO_3 + 0.5\ SiO_2 \tag{8}$$

J. Götze and M. Göbbels [23] have mentioned that an increase of pH-value above six increases the solubility of silica. EDS Mapping was used for analysis of this dried final product drawn from water solution, as shown in Figure 10.

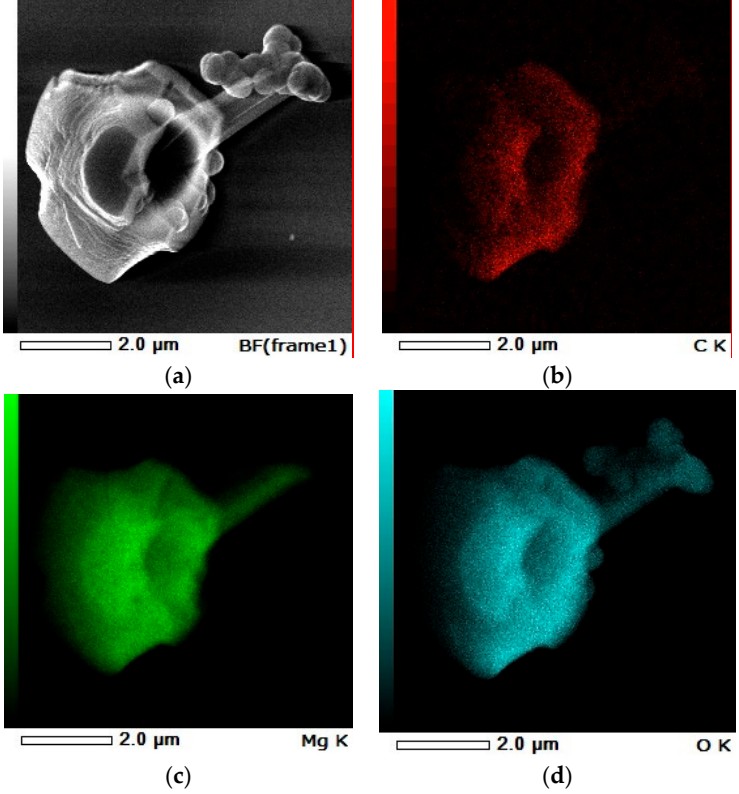

**Figure 10.** *Cont.*

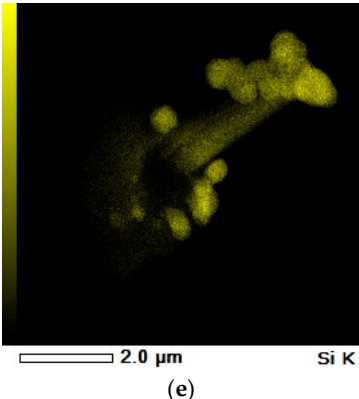

(**e**)

**Figure 10.** X-ray spectroscopy (EDS) mapping of final product obtained in the experiment V3F, (**a**) BF-SEM Image, (**b**) C K-carbon mapping, (**c**) Mg K-magnesium mapping, (**d**) O K-oxygen mapping, and (**e**) Si K-silicon mapping.

As shown in Figure 11, the TEM analysis of silicon oxide confirms that the formed silica particles have a spherical shape, with diameters of approximately 400–500 nm and the particles are amorphous.

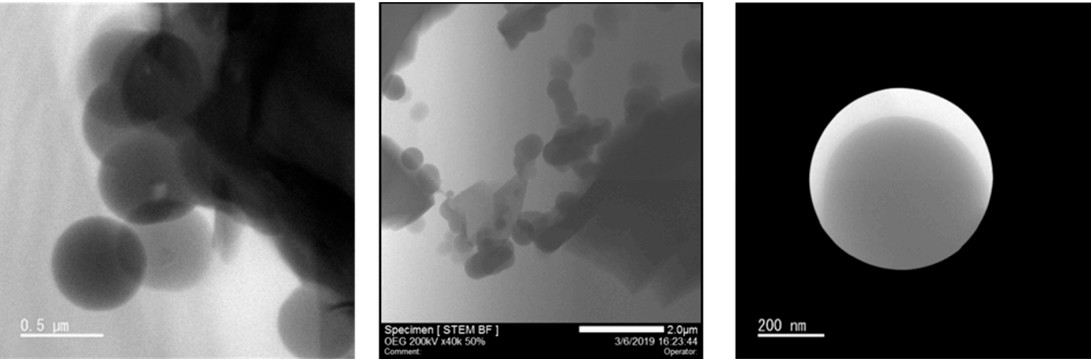

**Figure 11.** The TEM and STEM analysis of the obtained products.

Since an extensive share of the investigated material in Figure 10 shows low carbon content, there is a high possibility that this area is $Mg_2SiO_4$, what is previously shown in Figure 6. A new model for formation of nanosilica from magnesium silicate (forsterite) in used olivine was shown in Figure 12.

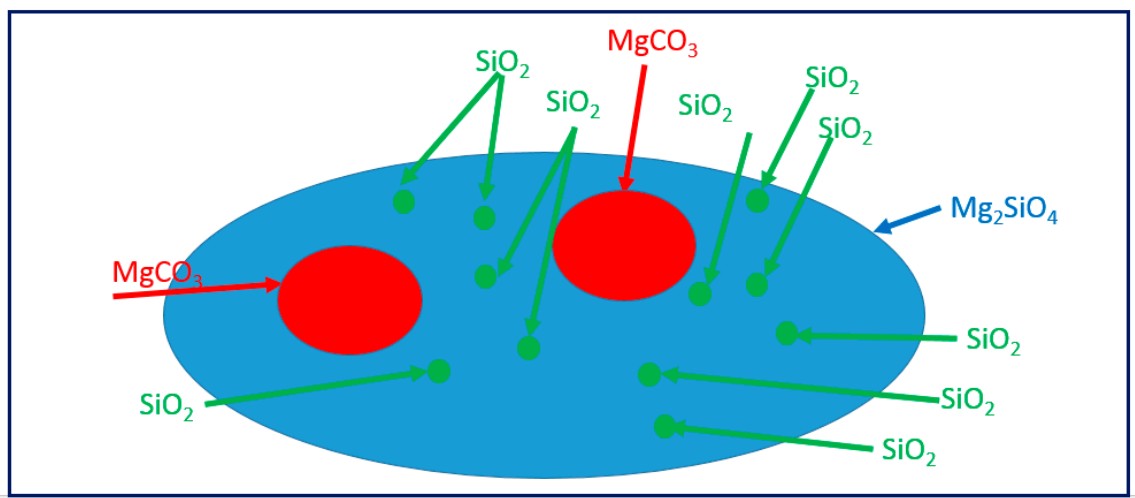

**Figure 12.** The proposed schematic model for the formation of $SiO_2$ from $Mg_2SiO_4$ in an autoclave.

We assumed that high stirring speed and high pressure in autoclave lead to the formation of ideally spherical silica particles together with larger fractions of magnesium carbonate. Weng et al. [24] found during a study of the kinetics and mechanism of mineral carbonation of olivine for $CO_2$ sequestration that the addition of sodium bicarbonate can dramatically increase the ionic strength and aid the dissolution of Si to temporarily aqueous $H_4SiO_4$ followed by decomposition to amorphous silica and consequently the removal of Si-rich layer. The aqueous silicon was not stable and can be decomposed into amorphous silica, which was extensively observed in the aqueous solution after carbonation and settled down for more than one month at room temperature.

### 3.3. Conclusions

Synthesis of nanosilica was studied via carbonation of olivine using size fraction between 20 and 63 μm with solid/liquid ratios of 1:10 at 175 °C and partial pressure of $CO_2$ more than 100 bar in an autoclave in the presence of additives such as sodium bicarbonate, oxalic and ascorbic acid. Under the above-mentioned conditions the ideally spherical particles of silica below 500 nm with amorphous grains were produced during carbonation. In comparison to ultrasonic spray pyrolysis, sol-gel and carbonation method via bubbling $CO_2$ gas into calcium ions dispersing in methanol solution under an atmospheric pressure, this synthesis was performed in a water solution in a closed reactor (autoclave) under higher pressure conditions above 100 bar avoiding the formation of silica gel, what blocks the metal extraction. An increase of stirring speed from 600 rpm to 1800 rpm raises the pressure from 120 to 170 bar and leads to an increase of silica production because of a removal of passivated silica formed layer at forsterite particles. The precipitated silicate particles were separated at pH-values between 8.32 and 8.57. In order to validate the first results in a 0.25 L-and 1.5 L autoclave, new scale up experiments will be performed in 10 L and 1000 L-autoclaves. Due to a good filterability of the carbonated product, separation of nanosilica from magnesite and magnesium carbonate shall be considered in future work in detail.

**Author Contributions:** S.S. conceptualized, managed the research, and co-wrote the paper. D.K. performed the preparation of the olivine materials (grinding, sieving) and co-wrote the paper. H.W. co-wrote the paper. C.D. participated in our experimental part, analyzed the data and co-wrote the paper. S.E. supervised the XRF- and XRD-analyses and co-wrote the paper with R.T. and B.F. supervised the personnel, provided funding and co-wrote the paper. I.K. performed the XRD, SEM, REM and STEM analysis of nanosilica. P.K. conceptualized the research and helped in the discussion of the morphological characteristics of obtained magnesium carbonate and nanosilica.

**Funding:** This research was funded by BMBF (Federal Ministry of Education and Research) in Berlin, grant number 033RCO14B (CO2MIN Project in period from 01.06.2017 to 31.05.2020).

**Acknowledgments:** For a continuous support and preparation of Figure 1 previously published in Metals in 2018, we would like to thank Andreas Bremen, AVT, RWTH Aachen University.

**Conflicts of Interest:** The authors declare no conflict of interest.

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
