# Peer review of "Synthesis of Nanosilica via Olivine Mineral Carbonation under High Pressure in an Autoclave"

_metals, doi:10.3390/met9060708_

Reviewer 1 Report

A valuable piece of research devoted to nanosilica production via olivine mineral carbonation. I liked the work , although with some small changes needed. 1 - Figure 7 is placed in the manuscript without a refer to it in the text. (the same case is valid for figure 10) 2 - Although. authors have mentioned "The challenge of future work is a separation of the formed nanosilica particles from the product" it is necessary to give implications on the filter-ability and feasibility of separation of the nanosilica particles from the feed stock and carbonate product. 3 - Based on figure 6, nucleation and growth of nanosilica particles occurs on the surface of the starting material. Is it technologically possible to have the nanosilica particles detached from the surface?

Author Response

Answer to Reviewer 1:

Dear reviewer,

I would like to thank for your very helpful comments.We added new Figure in our Text Figure 6: XRD analysis of Norwegian olivine samples; initial state (left) and carbonation product (right) in order to better explain the initial sample and product. Therefore Figure 7 becomes the Figure 8. We added “After filtration of the carbonated products, the white and yellow precipitate was appeared during staying after 7-10 days, as shown at Figure 8.”

 A valuable piece of research devoted to nanosilica production via olivine mineral carbonation. I liked the work , although with some small changes needed.

1 - Figure 7 is placed in the manuscript without a refer to it in the text. (the same case is valid for figure 10)

We added “After filtration of the carbonated products, the white and yellow precipitate was appeared during staying after 7-10 days, as shown at Figure 8.”

 As shown at Fig.11, the TEM analysis of silicon oxide confirms that the formed silica particles have a spherical shape, with diameters of approx. 400-500 nm and the particles are amorphous.

2 - Although. authors have mentioned "The challenge of future work is a separation of the formed nanosilica particles from the product" it is necessary to give implications on the filterability and feasibility of separation of the nanosilica particles from the feed stock and carbonate product.

We added: Because of a good filterability of the carbonated product, separation of nanosilica from magnesite and magnesium carbonate shall be considered in future work in detail..

3 - Based on figure 6, nucleation and growth of nanosilica particles occurs on the surface of the starting material. Is it technologically possible to have the nanosilica particles detached from the surface?

It is technologically possible to have the nanosilica detached from the surface using the small pellets from ZrO2 during the high pressure carbonation in an autoclave. This innovative idea was not tested, but it represents our new idea

Reviewer 2 Report

The submitted manuscript is an interesting account for the synthesis of SiO2 nanoparticles from natural olivine via decarbonation process. The approach is appropriate and the results justify the conclusions. I have only minor remarks that should be addressed prior to publication.
1. The Introduction section is quite long, although I presume that the authors just wanted to be precise in the method description. Specific comments:
- goethite is an iron-bearing hydroxide mineral of FeO(OH) chemical formula. Other formulas provided in the line 44 (HFeO2, Fe2O3.H2O) are just different ways to define the same crystal phase. They should be removed or the phrase could be rewritten in a clearer way;
- line 54 "(...) which contain both moderate and extensive SiO2 (...)" - did the authors mean rather moderate and extensive amounts of SiO2?
- line 79-81 "(...) effect of four parameters (boric acid, urea, feed rate of reactant, and reaction temperature) on particle size and standard deviation (...) line deviation decreased with increasing of all parameters except urea [11]". I would refrain from calling boric acid and urea "parameters", rather the concentrations or presence of these reagents can be named a parameter;
- Figure 1 could be provided in better resolution.
2. The origin of the natural sample of olivine was indicated as Norway, however more precise geographic details should be provided.
3. The semiquantitative XRD analysis results are reported in Table 2, but the details of the analysis procedure are missing. Please elaborate on how the XRD patterns were processed. In addition, the figures showing XRD patterns with the phase analysis should be included.
4. The Funding statement (lines 284-287) is a generic one, as provided in the template. Please correct it or remove the whole paragraph.

Author Response

Dear Reviewer,

Thank you very much for you very helpful comments. According to your comments, I am sending our answers:

1. The Introduction section is quite long, although I presume that the authors just wanted to be precise in the method description. Specific comments:
- goethite is an iron-bearing hydroxide mineral of FeO(OH) chemical formula. Other formulas provided in the line 44 (HFeO2, Fe2O3.H2O) are just different ways to define the same crystal phase. They should be removed or the phrase could be rewritten in a clearer way;

We removed the formulas from goethite and hydrated iron oxide from text.

- line 54 "(...) which contain both moderate and extensive SiO2 (...)" - did the authors mean rather moderate and extensive amounts of SiO2?

We changed it  in “rather moderate and extensive amounts of SiO2”
- line 79-81 "(...) effect of four parameters (boric acid, urea, feed rate of reactant, and reaction temperature) on particle size and standard deviation (...) line deviation decreased with increasing of all parameters except urea [11]". I would refrain from calling boric acid and urea "parameters", rather the concentrations or presence of these reagents can be named a parameter;

We are agreed, and improved it: “This work dealt with the effect of four parameters (the concentration of the boric acid and urea, feed rate of reactant, reaction temperature and time)” on particle size and standard deviation
-Figure 1 could be provided in better resolution.

I hope in the final version this Figure 1 will be provided in better resolution.
2. The origin of the natural sample of olivine was indicated as Norway, however more precise geographic details should be provided.
The samples used represent Steinsvik olivine from Norway
3. The semiquantitative XRD analysis results are reported in Table 2, but the details of the analysis procedure are missing. Please elaborate on how the XRD patterns were processed. In addition, the figures showing XRD patterns with the phase analysis should be included.

To evaluate the overall capability of the carbonation process, an experiment was performed using Norwegian olivine (20-63 µm) at 175°C, 120 bar, 120 min, 600 rpm, in the presence of additives of sodium carbonate, oxalic and ascorbic acid considering the present mineralogical phases detected via XRD before and after the carbonation (Figure 6)).
 Figure 6 was added in our Text.

Figure 6: XRD analysis of Norwegian olivine samples; initial state (left) and carbonation product (right)

4. The Funding statement (lines 284-287) is a generic one, as provided in the template. Please correct it or remove the whole paragraph.

Funding: This research was funded by BMBF (Federal Ministry of Education and Research) in Berlin, grant number 033RCO14B (CO2MIN Project in period from 01.06.2017 to 31.05.2020).